# Pre-clinical validation of a selective anti-cancer stem cell therapy for Numb-deficient human breast cancers

Daniela Tosoni[1], Sarah Pambianco[1], Blanche Ekalle Soppo[2], Silvia Zecchini[1], Giovanni Bertalot[1], Giancarlo Pruneri[1,3], Giuseppe Viale[1,3], Pier Paolo Di Fiore[1,2,3*,†] (ID) & Salvatore Pece[1,3,**,†] (ID)

## Abstract

The cell fate determinant Numb is frequently downregulated in human breast cancers (BCs), resulting in p53 inactivation and an aggressive disease course. In the mouse mammary gland, Numb/p53 downregulation leads to aberrant tissue morphogenesis, expansion of the stem cell compartment, and emergence of cancer stem cells (CSCs). Strikingly, CSC phenotypes in a Numb-knockout mouse model can be reverted by Numb/p53 restoration. Thus, targeting Numb/p53 dysfunction in Numb-deficient human BCs could represent a novel anti-CSC therapy. Here, using patient-derived xenografts, we show that expansion of the CSC pool, due to altered self-renewing divisions, is also a feature of Numb-deficient human BCs. In these cancers, using the inhibitor Nutlin-3 to restore p53, we corrected the defective self-renewal properties of Numb-deficient CSCs and inhibited CSC expansion, with a marked effect on tumorigenicity and metastasis. Remarkably, a regimen combining Nutlin-3 and chemotherapy induced persistent tumor growth inhibition, or even regression, and prevented CSC-driven tumor relapse after removal of chemotherapy. Our data provide a pre-clinical proof-of-concept that targeting Numb/p53 results in a specific anti-CSC therapy in human BCs.

**Keywords** breast cancer; cancer stem cells; Numb; p53; primary-derived xenografts

**Subject Categories** Cancer; Stem Cells

## Introduction

Cancer stem cells (CSC) have been identified in many types of cancers, including breast cancer (BC) (Reya *et al*, 2001; Al-Hajj *et al*, 2003; Dick, 2008; Shackleton *et al*, 2009; Clevers, 2011; Visvader, 2011; Beck & Blanpain, 2013; Kreso & Dick, 2014). In addition to their role in driving tumorigenesis, CSCs are thought to impact on therapy failure and disease progression, based on their intrinsic refractoriness to conventional anti-cancer therapies (Bao *et al*, 2006; Li *et al*, 2008; Dean, 2009; Diehn *et al*, 2009; Oravecz-Wilson *et al*, 2009; Liu & Wicha, 2010). Thus, the identification of anti-CSC therapies holds great promise to improve targeted cancer treatment. For such therapies to be developed, we need knowledge of the molecular mechanisms driving the emergence and maintenance of CSCs in the various cancer settings, and adequate pre-clinical models and assays for screening anti-CSC drugs. In this paper, we address these issues by providing: (i) evidence of a key molecular mechanism responsible for maintenance of CSCs in a subset of naturally occurring human BCs, and (ii) proof-of-concept of an effective anti-CSC therapy through the use of orthotopic patient-derived xenografts (PDX) and CSC-specific assays.

Previously, we demonstrated that the cell fate determinant Numb (Pece *et al*, 2011) is frequently downregulated in human BCs and that this event is associated with poor prognosis, suggestive of a tumor suppressor function of this protein (Pece *et al*, 2004; Colaluca *et al*, 2008). We further demonstrated, in a mouse model, that Numb exerts a dual homeostatic role in the mammary gland. In the stem cell (SC) compartment, Numb ensures the asymmetric outcome of self-renewing divisions by partitioning, at SC mitosis, into the progeny that retains the SC identity (Pece *et al*, 2010; Tosoni *et al*, 2015). In progenitors, the re-accumulation of Numb ensures proper maturation by preventing their reversion to a SC state (Tosoni *et al*, 2015). Accordingly, the selective ablation of Numb in the mouse mammary gland results in gross morphological alterations and development of pre-neoplastic lesions. These alterations are associated with expansion of the SC compartment and the emergence of cells displaying CSC traits, that is, a predominantly symmetric mode of self-renewing division, unlimited proliferation ability, epithelial-to-mesenchymal transition (EMT), and tumorigenicity upon orthotopic transplantation (Tosoni *et al*, 2015).

1   Istituto Europeo di Oncologia, Milan, Italy
2   IFOM, Fondazione Istituto FIRC di Oncologia Molecolare, Milan, Italy
3   Dipartimento di Oncologia e Emato-oncologia, Università degli Studi di Milano, Milan, Italy
   *Corresponding author. Tel: +39 02 574303257; E-mail: pierpaolo.difiore@ifom.eu
   **Corresponding author. Tel: +39 02 57489343; E-mail: salvatore.pece@ieo.it
   †These authors contributed equally to this work

At the mechanistic level, Numb inhibits the ubiquitination and ensuing degradation of p53 by the E3 ubiquitin ligase Mdm2 (Colaluca *et al*, 2008). Ablation of Numb therefore causes abnormal p53 degradation, which drives EMT and the conversion to the CSC state (Cicalese *et al*, 2009; Tosoni *et al*, 2015). Notably, the CSC traits acquired by Numb-ablated mouse mammary SCs are reverted by re-expression of Numb or restoration of p53 through the inhibition of Mdm2 with the small molecule inhibitor Nutlin-3 (Tosoni *et al*, 2015).

Based on these data and the frequent downregulation of Numb in human BCs, we hypothesized that, in Numb-deficient (Numb⁻) human BCs, the Numb/p53 dysfunction is causal in the emergence of CSCs, and that targeting this pathway will result in selective suppression of CSCs. Herein, we present the experimental validation of this hypothesis. To maximize the clinical relevance of our study, we employed orthotopic PDXs which, due to their ability to recapitulate the response of the parental tumors to treatments, represent more efficacious pre-clinical tools than the traditional, scarcely predictive models, such as established cell lines and their xenografts (DeRose *et al*, 2011; Gillet *et al*, 2011; Zhang *et al*, 2013; Wilding & Bodmer, 2014; Clohessy & Pandolfi, 2015; Whittle *et al*, 2015). Our results show that targeting the Numb/p53 pathway results in a CSC-specific therapy of Numb⁻ BCs.

# Results

### Characterization of human tumors and PDX models

To determine whether loss of Numb expression is critical for the maintenance of the CSC pool, we selected four Numb⁻ human BCs (T1–T4) displaying the typical clinical–pathological parameters that characterize this type of aggressive BC (Pece *et al*, 2004) (Fig 1A). Importantly, we verified a wild-type (WT) status of the *P53* gene in these tumors, meaning that a fully functional Numb/p53 pathway could in principle be restored (Fig 1A), and confirmed that loss of Numb expression was due to its increased proteasomal degradation (Fig 1B and Appendix Fig S1A), as described previously (Pece *et al*, 2004; Colaluca *et al*, 2008). To control that any effects we might observe were specific to loss of Numb expression and not merely associated with an aggressive BC phenotype *per se*, we also selected four Numb-proficient (Numb⁺) tumors (TA-TD) with clinical–pathological parameters similar to those of the Numb⁻ BCs (Fig 1A and B, and Appendix Fig S1A).

To study the tumorigenic properties of Numb⁻ CSCs, we developed clinically relevant PDX models, by xenotransplantation of fresh tumor explants into the inguinal mammary glands of immunocompromised mice. We confirmed by immunohistochemistry (IHC) on formalin-fixed, paraffin-embedded (FFPE) sections that the expression levels of Numb in the parental tumors were maintained in the corresponding PDXs (Fig 1C and Appendix Fig S1B).

### Numb-deficient human BCs display expansion of the CSC pool and altered SC self-renewal in an *in vitro* setting

In the mouse model, we demonstrated that Numb ablation results in expansion of the SC compartment (with characteristics of CSCs) due to an increased frequency of symmetric self-renewing divisions (Tosoni *et al*, 2015). To determine whether a similar scenario could be at play in Numb⁻ human BCs, we performed a series of assays *in vitro*, based on the mammosphere (MS) culture technique.

Mammosphere cultures were generated by culturing primary mammary epithelial cells (MECs), from the described tumors, in anchorage-independent conditions. Under such conditions, the majority of MECs undergo anoikis, while SCs survive and generate clonal spheroids or MSs (Dontu *et al*, 2004; Pece *et al*, 2010). We have previously established that MS cultures can be used, with due caution, as a proxy to assess the self-renewal properties of mammary SCs since: (i) the sphere forming efficiency (SFE) is indicative of the size of the SC pool (SC content); (ii) the serial propagation ability of MS cultures mirrors expansion of the SC pool (Pece *et al*, 2010; Tosoni *et al*, 2015).

We compared MSs generated from Numb⁻ BCs to MSs generated by normal MECs from the same patients (Fig 1D and E, and Appendix Fig S1C and D). Numb⁻ BC MSs were bigger and generated with a higher SFE, than the normal counterparts (Fig 1D and Appendix Fig S1C), and displayed continuous expansion upon serial propagation (Fig 1E and Appendix Fig S1D and E), probably as a consequence of symmetric self-renewal divisions performed by Numb⁻ MS-forming cells (MFC) (Appendix Fig S2A and B).

**Figure 1. Characterization of tumors and PDX models.**

A   Clinical and pathological features of breast tumors used in the study. Numb status was evaluated by IHC (see Materials and Methods). Tumor grade: G3, high; G2, intermediate. HER2 status was evaluated by IHC and confirmed by FISH (positive, POS; negative, NEG). Estrogen receptor (ER) and progesterone receptor (PR) status were evaluated by IHC and are reported as the percentage of positive cells. p53 wild-type (WT) status was established by sequencing of the p53 coding sequence (see Materials and Methods).

B   MSs derived from two Numb⁻ (T1 and T2) and two Numb⁺ (T3 and T4) tumors were treated *in vitro* with the proteasome inhibitor MG132 (0.5 μM for 48 h) and analyzed by IB as indicated. The increase in β-catenin was used as a control for the efficacy of proteasome inhibition by MG132. GRP94, loading control. Data for the other tumors (T3, T4, TC, TD) are in Appendix Fig S1A.

C   Top: Representative images of Numb IHC staining (brown) of hematoxylin–eosin counterstained FFPE sections from Numb⁻ (T1 and T2) and Numb⁺ (TA and TB) human primary BCs. Scale bar = 30 μm. Bottom: The four primary BCs were orthotopically xenografted into NGS mice, and the resulting PDXs were stained as in the top panel. Scale bar = 30 μm. Data for the other primary tumors (T3, T4, TC, TD) and corresponding PDXs are in Appendix Fig S1B.

D   Sphere forming efficiency (SFE) at passage 2 of the indicated MECs: N1 and T1, normal and tumor MECs from patient 1 (Numb⁻); N2 and T2, normal and tumor MECs from patient 2 (Numb⁻); TA and TB, tumor MECs form patients A and B (Numb⁺). For each tumor, data are expressed as the mean of four independent experiments (± SD of 12 measurements).

E   A typical serial propagation experiment with MSs obtained from MECs as in (D). The cumulative sphere number over four passages is reported. See also Appendix Fig S1D for a detailed description of the serial propagation assay exemplified using N1 and T1 samples. Shown data are from experiments representative of three biological replicas and are expressed as the mean value of technical triplicates. When not indicated, SD was < 30% of the mean. Data for the other tumors (T3, T4, TC, TD) are in Appendix Fig S1E.

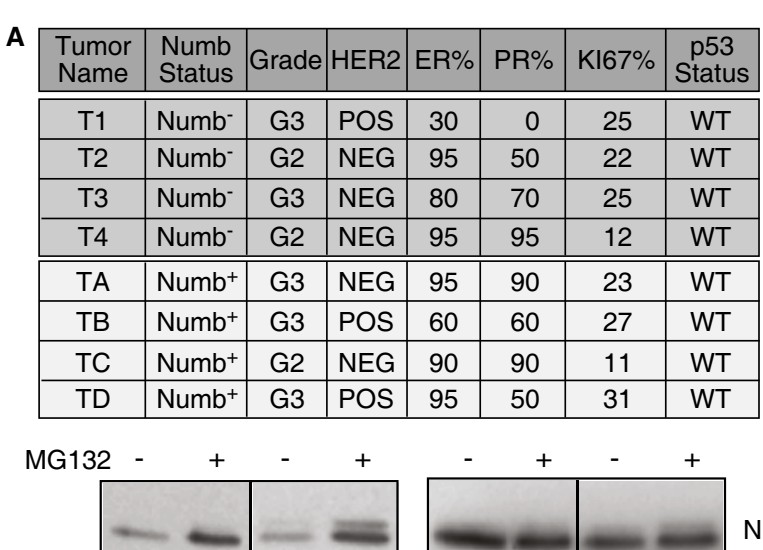

**A**

| Tumor Name | Numb Status | Grade | HER2 | ER% | PR% | KI67% | p53 Status |
|---|---|---|---|---|---|---|---|
| T1 | Numb⁻ | G3 | POS | 30 | 0 | 25 | WT |
| T2 | Numb⁻ | G2 | NEG | 95 | 50 | 22 | WT |
| T3 | Numb⁻ | G3 | NEG | 80 | 70 | 25 | WT |
| T4 | Numb⁻ | G2 | NEG | 95 | 95 | 12 | WT |
| TA | Numb⁺ | G3 | NEG | 95 | 90 | 23 | WT |
| TB | Numb⁺ | G3 | POS | 60 | 60 | 27 | WT |
| TC | Numb⁺ | G2 | NEG | 90 | 90 | 11 | WT |
| TD | Numb⁺ | G3 | POS | 95 | 50 | 31 | WT |

**B**

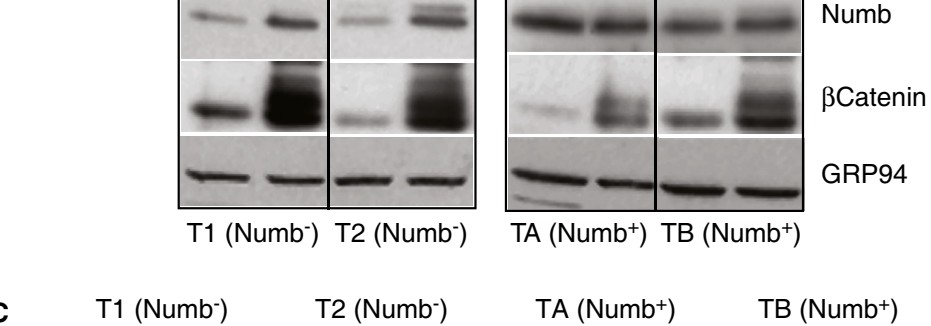

**C**

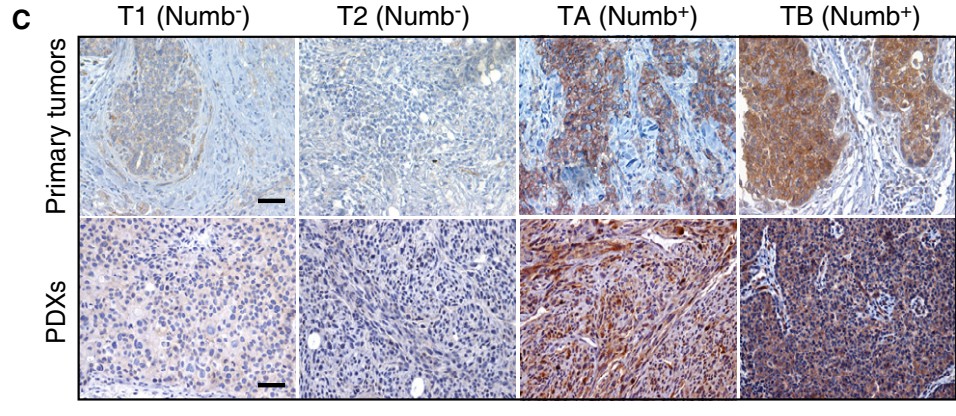

**D**

| Sample | SFE (%) |
|---|---|
| N1 | 0.12 ± 0.04 |
| N2 | 0.13 ± 0.03 |
| T1 | 0.47 ± 0.08 |
| T2 | 0.48 ± 0.08 |
| TA | 0.50 ± 0.07 |
| TB | 0.44 ± 0.06 |

**E**

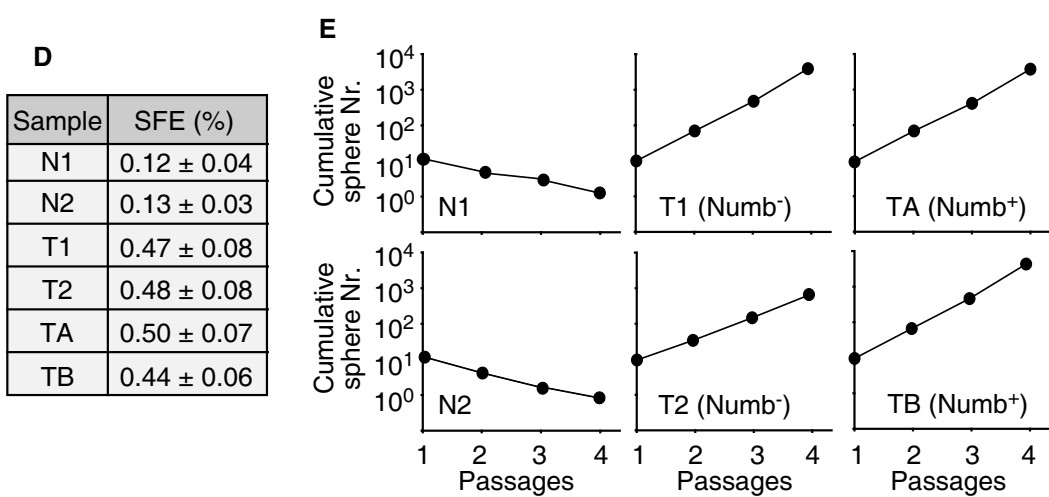

Figure 1.

Conversely, normal MSs displayed a progressive decrease in self-renewing potential (Fig 1E and Appendix Fig S1D) and the MFCs performed asymmetric self-renewal divisions, as also witnessed by the asymmetric partitioning of Numb at mitosis of the MFC (Appendix Fig S2C and D). Thus, the properties of Numb⁻ BC MSs are consistent with an expanded CSC pool with increased self-renewal potential.

Finally, we assessed the serial propagation ability of MSs derived from the four Numb⁺ human BCs (TA-D) selected as controls. For these tumors to be suitable controls that would permit us to attribute specific effects to a Numb⁻ status rather than to an aggressive tumor phenotype, it was necessary that they displayed similar properties, in the MS assay, to those of Numb⁻ tumors. We verified that this was indeed the case by analyzing the SFE (Fig 1D) and the kinetics of MS expansion in the serial propagation assay (Fig 1E and Appendix Fig S1E and F), which were indeed comparable between the analyzed Numb⁻ and Numb⁺ BCs. The observation that Numb⁺ human breast CSCs also display increased self-renewal potential highlights the existence of alternative Numb-independent mechanisms responsible for SC expansion in high-grade BCs.

### Loss of Numb is responsible for expansion of the SC compartment in Numb⁻ BCs

The above data are consistent with the notion that loss of Numb expression in Numb⁻ BCs causes expansion of the SC compartment. To test this possibility, we re-expressed a DsRed-conjugated Numb protein (Numb-DsRed) in tumor MECs (Fig 2A), and tested its effects *in vitro* and *in vivo*. As expected, re-expression of Numb in Numb⁻ MECs resulted in increased p53 levels (Fig 2A) and function, determined by qPCR of p53 target genes (Fig 2B), while p53 in Numb⁺ MECs was unaffected (Fig 2A and B).

In an *in vitro* setting, re-expression of Numb in Numb⁻ MECs caused: (i) a reduced ability to generate MS, as evidenced by a reduction in SFE and average MS size (Fig 2C); (ii) a switch in the mode of the first mitotic division of the MFC (Fig 2D and Appendix Fig S2B), from symmetric to asymmetric, as assessed by time-lapse video microscopy; (iii) a marked reduction in the self-renewal capacity of the SC population in the serial MS propagation assay (Fig 2E). In contrast, the expression of DsRed-Numb in Numb⁺ MECs had no effect on the self-renewal properties of CSCs (Fig 2C–E).

In the PDX model system, Numb-reconstituted Numb⁻ MECs generated tumors that were reduced in size by ~40% compared with mock-infected controls, while Numb overexpression had no effect on the growth of tumors generated by Numb⁺ MECs (Fig 3A). Tumors generated from Numb-reconstituted Numb⁻ MECs showed no evidence of increased apoptosis or decreased proliferation in the bulk tumor population (Fig 3B and C), arguing in favor of a selective effect of Numb restoration on the CSC compartment. To prove this, we directly measured the number of cancer-initiating cells (CIC) (functionally equivalent to CSCs) by limiting dilution orthotopic transplantation. We observed that Numb restoration in Numb⁻ MECs resulted in a 70–75% reduction in CICs, but had no effect on Numb⁺ MECs (Table 1).

These results argue that, in Numb⁻ tumors, loss of Numb expression causes the expansion of the SC compartment leading to tumorigenesis. Although the mechanisms responsible for SC expansion in Numb⁺ tumors remain to be established, our data show that these tumors represent a suitable control for the Numb-dependent phenotypes observed in Numb⁻ tumors.

### Attenuation of p53 activity drives expansion of the SC compartment in Numb⁻ tumors

We previously demonstrated, in a mouse model, that the asymmetric partitioning of Numb during mitotic division of normal mammary SCs determines asymmetric cell division (Tosoni *et al*, 2015). The downstream effector in this pathway is p53 (Tosoni *et al*, 2015), whose levels are stabilized due to the inhibition of Mdm2 by Numb (Colaluca *et al*, 2008).

We reasoned that the subversion of this mechanism might be responsible, at least in part, for the alterations in the SC compartment in Numb⁻ human BCs. If so, restoration of p53 activity should attenuate the CSC phenotypes caused by loss of Numb. To restore p53 activity, we used Nutlin-3, a small molecule inhibitor that prevents Mdm2-mediated ubiquitination of p53 (Vassilev *et al*, 2004).

Figure 2.  Effects of Numb re-expression on the self-renewal properties of CSCs derived from Numb⁻ and Numb⁺ BCs.

A   MSs derived from Numb⁻ and Numb⁺ BCs were transduced with Numb-DsRed and analyzed by IB to verify levels of Numb, p53, and the p53 target gene p21. Black arrow, endogenous Numb; red arrow, Numb-DsRed; GRP94, loading control.

B   MSs derived from the indicated BCs and transduced as in (A) were analyzed by qPCR for the expression of p53 target genes: *CDKN1A* (p21) and *MDM2*, positively regulated by p53, *NANOG*, repressed by p53. +Nb, reconstituted with Numb-DsRed; Ctr, mock-infected. Data are from a single experiment.

C   MSs obtained from the indicated BCs were reconstituted with Numb-DsRed (+Nb) or mock-infected (Ctr) and assessed for SFE and MS size. Data are mean values of three independent experiments (± SD of 12 measurements) and are expressed relative to the SFE or MS size in Ctr cells (= 100%). Unpaired two-sided Student's *t*-test. *P < 0.05 vs. Ctr in the matching condition was considered as significant (P-value SFE: T1, 3.06E-07; T2, 6.93E-07; T3, 1.18E-07; T4, 4.73E-09; P-value MS size: T1, 0.0093; T2, 8E-19; T3, 2.8E-21; T4, 7.7E-09).

D   MSs treated as in (C) were assessed for SC mode of division. The percentage of asymmetric and symmetric divisions for each condition (Numb⁻ and Numb⁺ SCs; Ctr and Numb-dsRed-reconstituted SCs) was calculated by following the pattern of cell number progression within the forming MS (1-2-3-5 progression = asymmetric division of the SC; 1-2-4-6/8 progression = symmetric division of the SC; see also Appendix Fig S2 for a detailed description of the criteria). For each condition, the bars represent the mean value, from three independent experiments, of the percentage of asymmetric (Asym.) vs. symmetric divisions (Sym.) performed by the MS-forming cell (MFC). The SD was calculated from a minimum of 50 MFCs for each condition. Non-parametric Fisher's exact test. *P < 0.05 vs. Ctr in the matching condition was considered as significant (P-value T1, 4.03E-07; T2, 8.91E-04).

E   MSs treated as in (C) were assessed for self-renewal ability in the serial MS propagation assay. Shown data are from experiments representative of three biological replicas and are expressed as the mean value of technical triplicates. When not indicated, SD was < 30% of the mean. Linear regression analysis followed by unpaired two-sided Student's *t*-test was used to compare the slopes of the regression lines. P < 0.05 vs. Ctr in the matching condition was considered as significant (P-values: T1, 2.02E-05; T2, 3.45E-05; T3, 9.17E-06; T4, 7.74E-05). See also Appendix Figs S1 and S2 for additional experimental and methodological details.

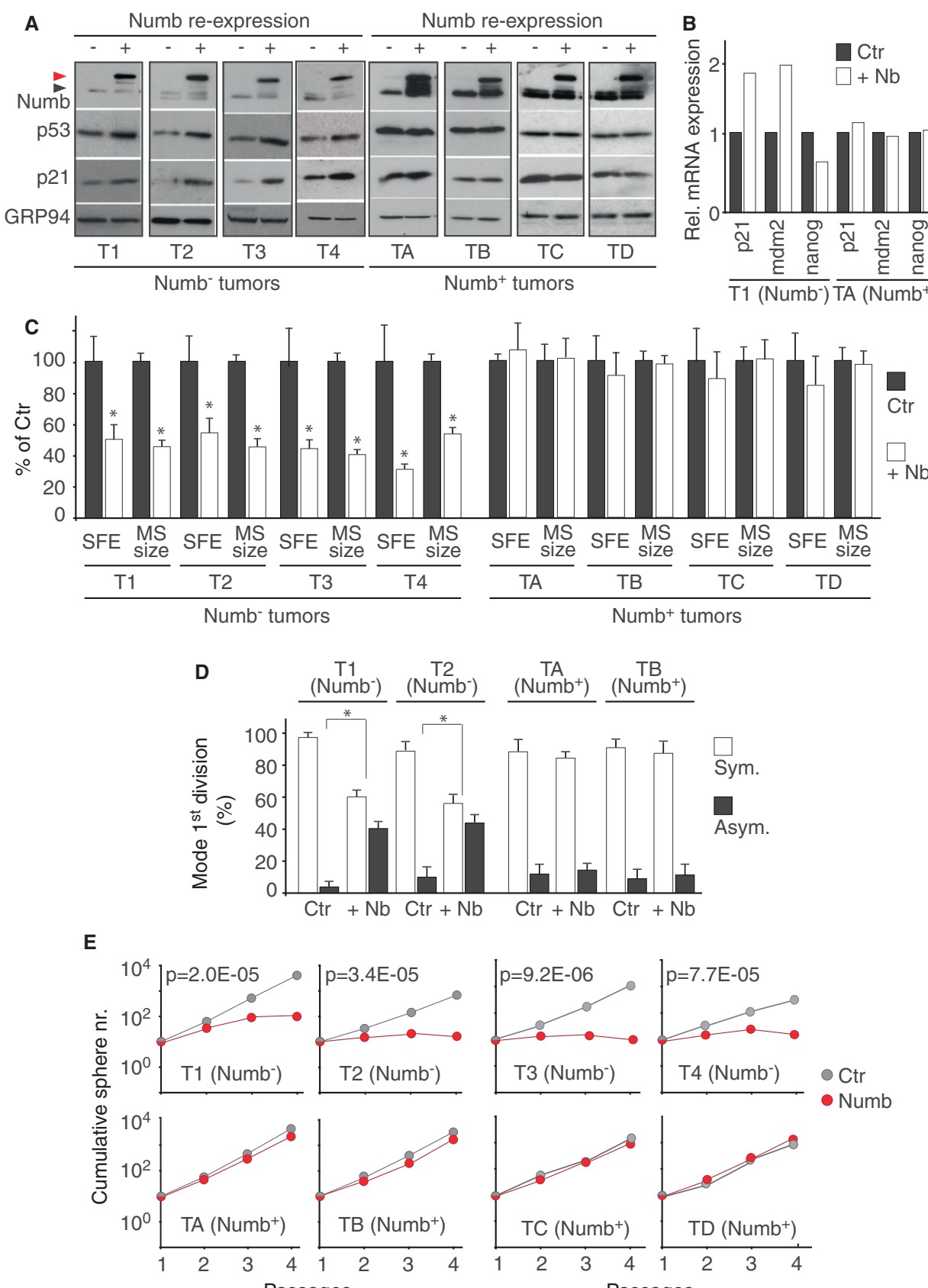

**Figure 2.**

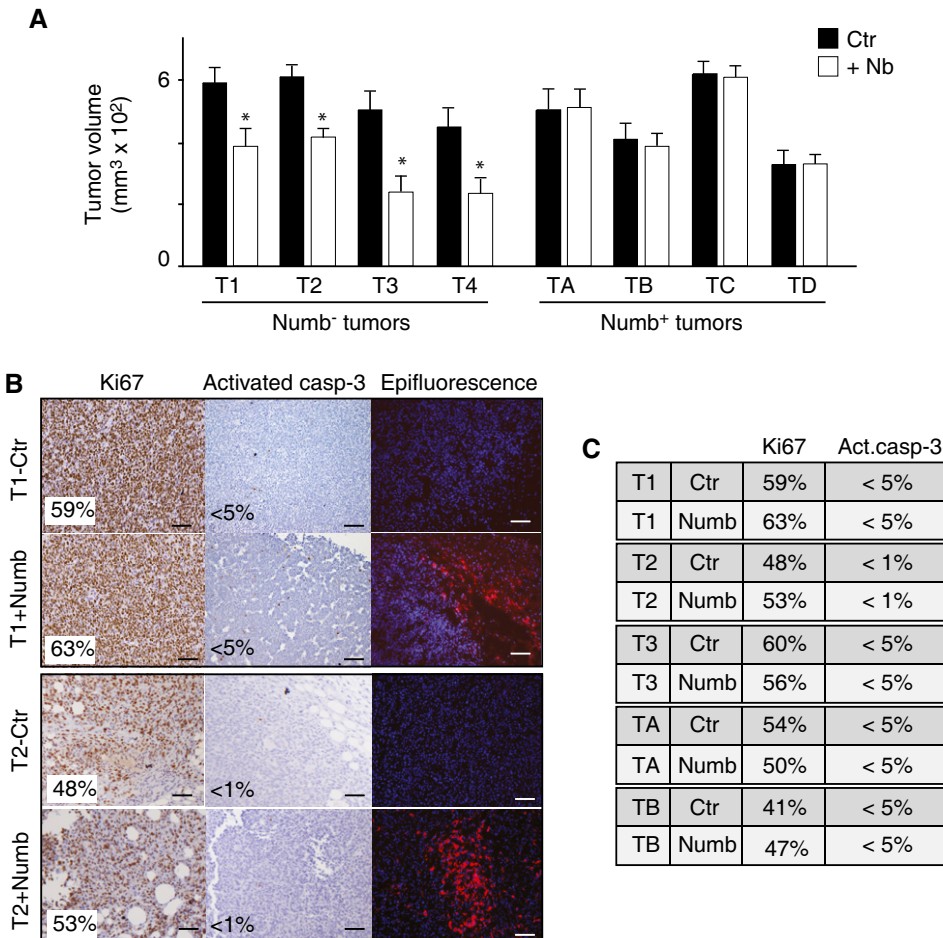

**Figure 3.  Effects of Numb re-expression on tumorigenicity and CSC content of Numb⁻ and Numb⁺ tumors.**

A    Numb⁻ and Numb⁺ MECs, reconstituted with Numb-DsRed (+Nb) or mock-infected (Ctr), were orthotopically transplanted (50,000 cells per injection) in NGS mice. Animals were sacrificed at 15–21 days post-injection, and tumor volume was assessed by caliper measurement immediately after excision. Bars represent the tumor volume, expressed as the mean value of three independent experiments ($\pm$ SD of a minimum of 11 to a maximum of 16 tumors). Unpaired two-sided Student's $t$-test. *$P < 0.05$ vs. Ctr in the matching condition was considered as significant (T1, $P = 1.65E-10$; T2, $P = 6.9E-16$; T3, $P = 1.13E-12$; T4, $P = 3.95E-08$).

B, C    IHC analysis of Ki67 and activated caspase-3 in tumors generated from control (Ctr) or Numb-DsRed (+Numb) cells. Representative IHC stainings are shown in (B), while in (C) the quantification of the IHC analysis of all tumors is reported (> 30,000 cells counted). The "Epifluorescence" panel in (B) shows the expression of Numb-DsRed in the tumors. Scale bars = 100 μm.

Treatment with Nutlin-3 caused an increase in p53 levels in Numb⁻, but not in Numb⁺, MECs (Fig 4A). This was accompanied by enhanced p53 activity, as witnessed by increased levels of p53 target genes (Fig 4B). At the biological level, Nutlin-3 induced in Numb⁻, but not Numb⁺, MECs: (i) a reduction in SFE and MS size (Fig 4C); (ii) a reduction in the ability of MSs to be serially propagated (Fig 4D); (iii) a switch in the mode of division from symmetric to asymmetric (Fig 4E). At the dose employed (10 μM), Nutlin-3 did not exert significant toxic effects, as assessed by monitoring the levels of proliferation (Ki67) and apoptosis (activated caspase-3) markers (Appendix Fig S3A). In addition, the effects of Nutlin-3 were unlikely due to off-target effects of the drug, since an inactive enantiomer, Nutlin-3b, was inactive in all experiments at concentrations five times higher than those used for Nutlin-3 (Appendix Fig S3A and B).

We concluded that attenuation of p53 activity in Numb⁻ BCs is directly responsible for de-regulation of SC self-renewal and expansion of the SC compartment.

**Pharmacological restoration of p53 selectively targets the CSC population in PDXs**

Based on the above results, we hypothesized that pharmacological targeting of the Numb/p53 axis might represent an effective anti-CSC therapy in Numb⁻ BCs. To validate this concept in a pre-clinical setting that closely recapitulates real tumors, we took advantage of the PDX models.

In initial experiments, we pre-treated Numb⁻ and Numb⁺ MECs *in vitro* with Nutlin-3 and then orthotopically xenografted them. Treated Numb⁻ (but not Numb⁺) MECs generated PDXs with a significantly slower growth rate compared to those derived from mock-treated cells (Fig 4F). This effect occurred in the absence of detectable modifications in the rate of proliferation or apoptosis in the bulk tumor cell population (Fig 4G). Controls performed with the inactive enantiomer, Nutlin-3b, showed that the effect of Nutlin-3 was specific (Appendix Fig S3C). Furthermore, in a tail vein

**Table 1. Numb re-expression affects the tumorigenicity and CSC content of Numb⁻ tumors, but not of Numb⁺ tumors.**

| | Tumor | Treatment | Nr. of cells injected | | | | | | Freq. SCs/CICs | 95% CI | *P*-value |
|---|---|---|---|---|---|---|---|---|---|---|---|
| | | | $10^5$ | $10^4$ | $5 \times 10^3$ | $10^3$ | $10^2$ | $10^1$ | | | |
| | | | Outgrowths/injections | | | | | | | | |
| Numb⁻ tumors | T3 | Ctr | 2/2 | 4/4 | 3/4 | 2/8 | 0/8 | 0/4 | 1:3,290 | 1:1,550–1:6,985 | 0.013 |
| | T3 | Numb | 2/2 | 3/6 | 2/6 | 0/2 | 0/2 | 0/2 | 1:13,999 | 1:5,803–1:33,772 | |
| | T4 | Ctr | 2/2 | 4/4 | 3/4 | 1/8 | 0/8 | 0/4 | 1:3,954 | 1:1,857–1:8,420 | 0.017 |
| | T4 | Numb | 2/2 | 4/8 | 2/8 | 0/2 | 0/2 | 0/2 | 1:15,751 | 1:7,110–1:34,892 | |
| Numb⁺ tumors | TC | Ctr | 2/2 | 2/2 | 5/7 | 0/8 | 0/2 | 0/2 | 1:5,090 | 1:2,396–1:10,811 | 0.949 |
| | TC | Numb | 2/2 | 2/2 | 3/5 | 1/6 | 0/8 | 0/4 | 1:4,897 | 1:2,116–1:11,335 | |
| | TD | Ctr | 2/2 | 2/2 | 4/6 | 1/8 | 0/2 | 0/2 | 1:4,660 | 1:2,147–1:10,115 | 0.649 |
| | TD | Numb | 2/2 | 2/2 | 3/6 | 1/8 | 0/2 | 0/2 | 1:6,096 | 1:2,685–1:13,839 | |

Numb⁻ or Numb⁺ MECs, reconstituted with Numb-DsRed (Numb) or mock-infected (Ctr), were orthotopically transplanted at limiting dilutions in NGS mice. The number of outgrowths per number of injected pads (Outgrowths/injections) is indicated. The complete statistical analysis of the experiment, with the frequency of CICs and with 95% confidence intervals (CI), is shown. The frequencies were calculated by Poisson statistics, using the "StatMod" software package for the R computing environment (http://www.R-project.org), as previously described (Shackleton *et al*, 2006), and a complementary log-log generalized linear model (two-sided 95% Wald confidence intervals or, in case of zero outgrowths, one-sided 95% Clopper–Pearson intervals). The single-hit assumption was tested as recommended and was not rejected for any dilution series ($P > 0.05$).

injection model, *in vitro* pre-treatment with Nutlin-3 significantly suppressed the colonization of the lung by Numb⁻ (but not Numb⁺) MECs (Fig 5).

We then moved to a more stringent setting, based on the *in vivo* Nutlin-3 treatment of PDXs that had reached palpable size (~20 mm³). At the end of the *in vivo* treatment (12 days), no effect on tumor size was observed (Fig 6A), despite increases in p53 levels caused by Nutlin-3 (Fig 6B). However, MECs derived from Numb⁻ (but not Numb⁺) primary PDXs that had been treated with Nutlin-3 *in vivo* displayed a significantly reduced MS-SFE *in vitro* compared with controls (Fig 6C, left). Moreover, upon re-transplantation, performed with no further drug treatment, these cells generated tumors that were significantly smaller than their corresponding controls (Fig 6C, right), in the absence of any detectable effect on

the rate of proliferation or apoptosis in the bulk tumor mass (Fig 6D).

The observation that *in vivo* treatment with Nutlin-3 did not influence the growth of the primary PDXs (at least in the timeframe of our experiment), but affected the SFE *in vitro* and the growth of second generation PDXs, is consistent with the notion that Nutlin-3 is a selective anti-CSC therapy in Numb⁻ BCs. Indeed, one would predict that a therapy selectively targeting slow-proliferating CSCs, while having no effect on actively dividing progenitors, would have little effect in the short-term on tumor growth, but instead become effective once the progenitors have exhausted their replicative ability.

We sought direct proof for this possibility by performing limiting dilution xenotransplantation experiments. PDXs were grown until

**Figure 4. Effects of Nutlin-3 on the mode of division of Numb⁻ and Numb⁺ tumors.**

A, B  MSs derived from Numb⁻ and Numb⁺ tumors were treated with 10 μM Nutlin-3 and analyzed by IB (A) or qPCR (B) to evaluate the restoration of p53. In (A), GRP94, loading control. Panels shown in the IB were assembled either using lanes from the same blot (splicing out lanes loaded with additional controls or non-relevant samples), or from different blots run and stained simultaneously (see also Materials and Methods). In (B), *CDKN1A* (p21) and *MDM2*, positively regulated p53 target genes. Data are from a single experiment.

C–F  MECs derived from the indicated BCs were treated *in vitro* with 10 μM Nutlin-3 or vehicle (Ctr) and assessed for SFE and MS size (C), MS serial propagation ability (D), mode of division (E), and tumorigenicity (F). In (C), data are mean values of three independent experiments (± SD of 12 measurements) and are expressed relative to the SFE or MS size in Ctr cells (= 100%). Unpaired two-sided Student's *t*-test. *$P < 0.05$ vs. Ctr in the matching condition was considered as significant (*P*-value SFE: T1, 5.65E-07; T2, 2.38E-08, *P*-value MS size: T1, 4.99E-12; T2, 6.88E-14). In (D), data are from experiments representative of three biological replicas and are expressed as the mean value of technical triplicates. When not indicated, SD was < 30% of the mean. Linear regression analysis followed by unpaired two-sided Student's *t*-test. $P < 0.05$ vs. Ctr in the matching condition was considered as significant (*P*-values: T1, 5.03E-06; T2, 1.83E-05). In (E), the percentage of asymmetric (Asym.) and symmetric (Sym.) divisions for each condition (Numb⁻ and Numb⁺ SCs; Ctr and Nutlin-treated SCs) was calculated based on the pattern of asymmetric vs. symmetric division (see also legends to Fig 2D and Appendix Fig S2 for further details). For each condition, the bars represent the mean value, from three independent experiments, of the percentage of asymmetric (Asym.) vs. symmetric divisions (Sym.) performed by the MS-forming cell (MFC). The SD was calculated from a minimum of 50 MFCs for each condition. Non-parametric Fisher's exact test. *$P < 0.05$ vs. Ctr in the matching condition was considered as significant (*P*-value T1, 9.62E-11; T2, 6.60E-03). In (F), Numb⁻ and Numb⁺ MECs, treated with Nutlin or vehicle (Ctr), were orthotopically transplanted (50,000 cells per injection) in NGS mice. Animals were sacrificed at 15–21 days post-injection, and tumor volume was assessed by caliper measurement immediately after excision. Bars represent the tumor volume, expressed as the mean value of three independent experiments (± SD of a minimum of 11 to a maximum of 16 tumors). Unpaired two-sided Student's *t*-test. *$P < 0.05$ vs. Ctr in the matching condition was considered as significant (T1, $P$ = 5.09E-18; T2, $P$ = 2E-17).

G  IHC analysis of Ki67 and activated caspase-3 expression in tumors obtained from orthotopic transplantation of human BC MECs (T1 and T2 Numb⁻ tumors) pre-treated *in vitro* with Nutlin-3. The percentage of positive cells is shown in the panels (> 30,000 cells counted/sample). Scale bars = 100 μm.

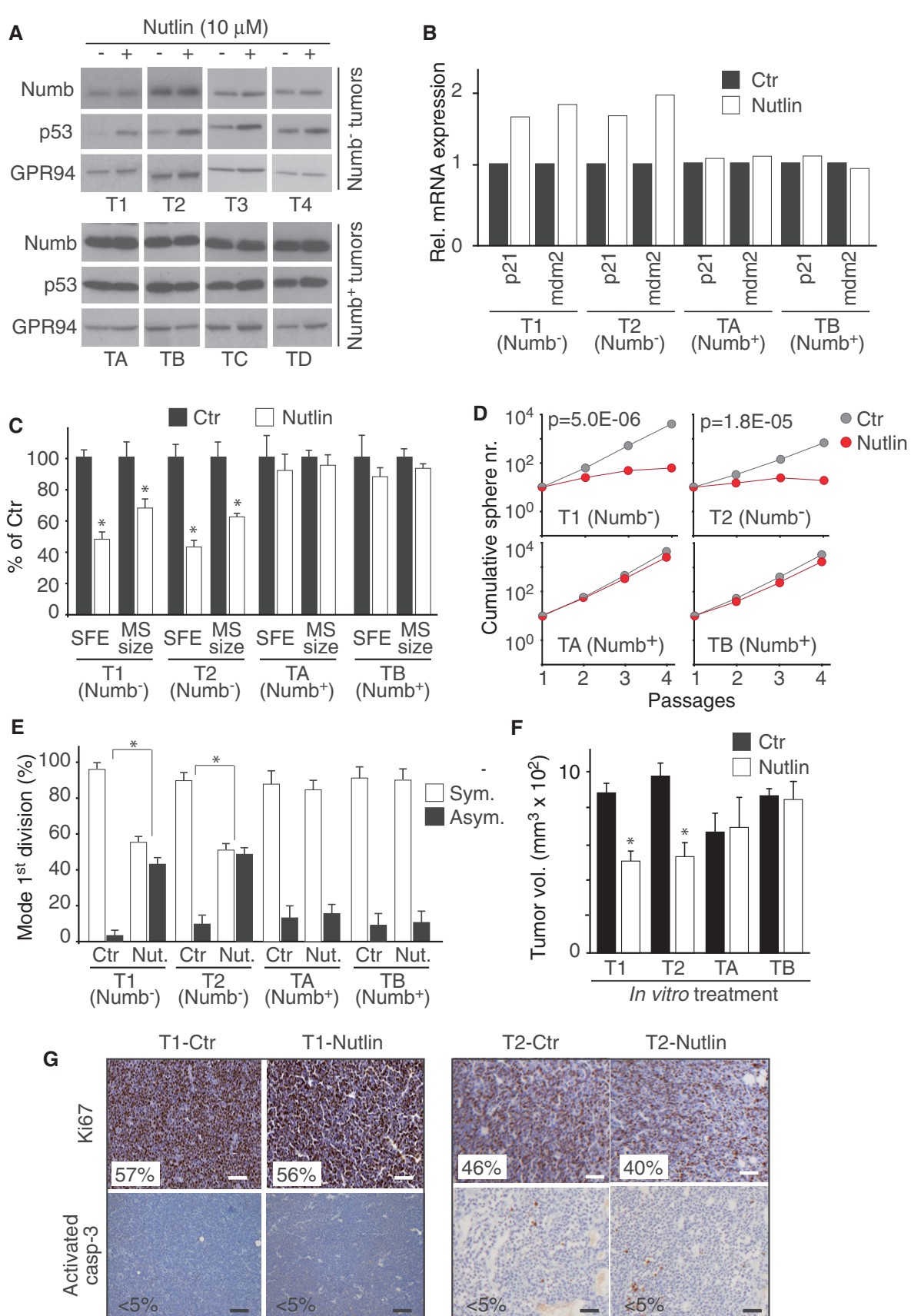

**Figure 4.**

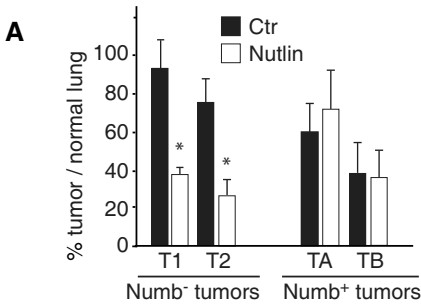

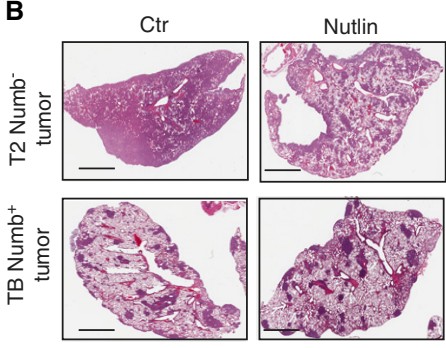

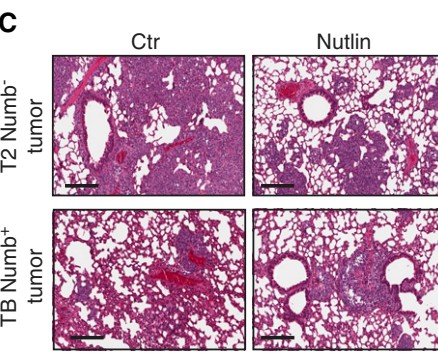

**Figure 5. Effects of Nutlin-3 treatment on metastasis of Numb⁻ and Numb⁺ tumors *in vivo*.**

A–C  MSs derived from T1 and T2 Numb⁻ and TA and TB Numb⁺ tumors were treated with 10 μM Nutlin-3, or vehicle as a control, and injected into the lateral tail vein of NSG mice (250,000 cells/mouse; 4 mice for each treatment group). The metastatic burden in the lung was analyzed 3 weeks post-injection. In (A), bars represent the percentage of area occupied by metastatic nodules with respect to area of normal lung tissue. For each treatment group, data and error bars show mean ± SD calculated in 10 randomly selected optical fields analyzed in histological sections of metastasis from four representative animals (two Numb⁻ and two Numb⁺ tumors). Unpaired two-sided Student's *t*-test. *$P$ < 0.05 vs. vehicle in the matching condition was considered as significant (T1, $P$ = 1.14E-05; T2, $P$ = 1.78E-05). In (B), representative histological images of lungs obtained from mice injected with T2 Numb⁻ and TB Numb⁺ cells. Scale bars = 2 mm. In (C), magnifications of the IHC sections reported in (B). Scale bars = 150 μm.

palpable (~20 mm³) before commencing with Nutlin-3 treatment. After 12 days of treatment, PDX-derived cells were orthotopically re-transplanted, in the absence of any further exposure to the drug, at limiting dilutions. We observed a pronounced decrease (5–6-fold)

in the frequency of CICs in Nutlin-3-treated Numb⁻ PDXs compared with controls, while no effect was observed on Numb⁺ PDXs (Table 2).

## Nutlin-3 potentiates the anti-tumoral effects of chemotherapy and prevents tumor growth relapse after chemotherapy discontinuation

Based on the emerging notion that CSCs are able to drive tumor recurrence after initial transient response to chemotherapy, we used the PDX models to test the efficacy of Nutlin-3 in combination with paclitaxel, a standard-of-care chemotherapy drug used in many types of cancer, including breast cancer (Goldhirsch *et al*, 2007). To generate a sufficient number of tumor-bearing mice to be randomized to treatment with Nutlin-3 and paclitaxel either alone or in combination (see Materials and Methods), we selected three additional Numb⁻ PDXs and one Numb⁺ PDX as a control (characterized in Appendix Fig S4A and B). MECs from these PDXs were transplanted into NOD/SCID mice and allowed to form tumors with a minimal size of ~100–200 mm³ to better monitor tumor burden variations in response to treatments. In keeping with our previous observations (see Fig 6A), Nutlin-3 showed no effect on the initial tumor response (Fig 7, left; day 15), regardless of the Numb status. Tumor-bearing mice belonging to the control (vehicle only) and Nutlin-3 groups had to be invariably removed from the study within 15 days of the start of the treatment due to excessive tumor growth (Fig 7, left). In contrast, paclitaxel was highly efficacious in causing initial inhibition of tumor growth accompanied by anti-proliferative and/or pro-apoptotic effects (Appendix Fig S4C and D). However, upon therapy discontinuation all tumors, regardless of the Numb status, displayed robust re-growth, compatible with the existence of a paclitaxel-resistant subpopulation (Fig 7, left).

Remarkably, Nutlin-3 increased the response to paclitaxel, during the treatment period, with a synergistic effect ranging from an ~30% increase in tumor reduction to complete tumor regression (Fig 7, left). More importantly, in the Nutlin-3 + paclitaxel group of Numb⁻ (but not in the Numb⁺) PDXs, we never observed tumor re-growth. In this subgroup, by the end of the observation period, animals showed either a significantly decreased tumor burden compared to baseline tumors (a measure equivalent to the "objective response rate, ORR" used in clinical trials), or complete tumor regression (equivalent to a "pathological complete response, pCR" in the clinical setting) (T7 Numb⁻ tumor in Fig 7, left).

To evaluate the *in vivo* effects of the different treatments on the CSC fraction, MECs were isolated from tumors of each group (at the end of the treatment period *in vivo*) and tested, in the absence of any further treatment, for MS formation efficiency *in vitro* and tumorigenicity *in vivo*, or analyzed by flow cytometry for the expression of surface markers previously linked to a breast CSC phenotype (Al-Hajj *et al*, 2003; Ginestier *et al*, 2007; Ricardo *et al*, 2011). Numb⁻ MECs derived from PDXs treated *in vivo* with Nutlin-3, either alone or in combination with paclitaxel, displayed significant reduction in SFE and tumor formation (Fig 7, middle and right), accompanied by decreased ALDH activity or reduced number of CD44⁺/CD24⁻ cells (Appendix Fig S5). Conversely, no effects were evident in Numb⁻ MECs obtained from the paclitaxel only group or in MECs from the Numb⁺ PDX (Fig 7, middle and right; Appendix Fig S5).

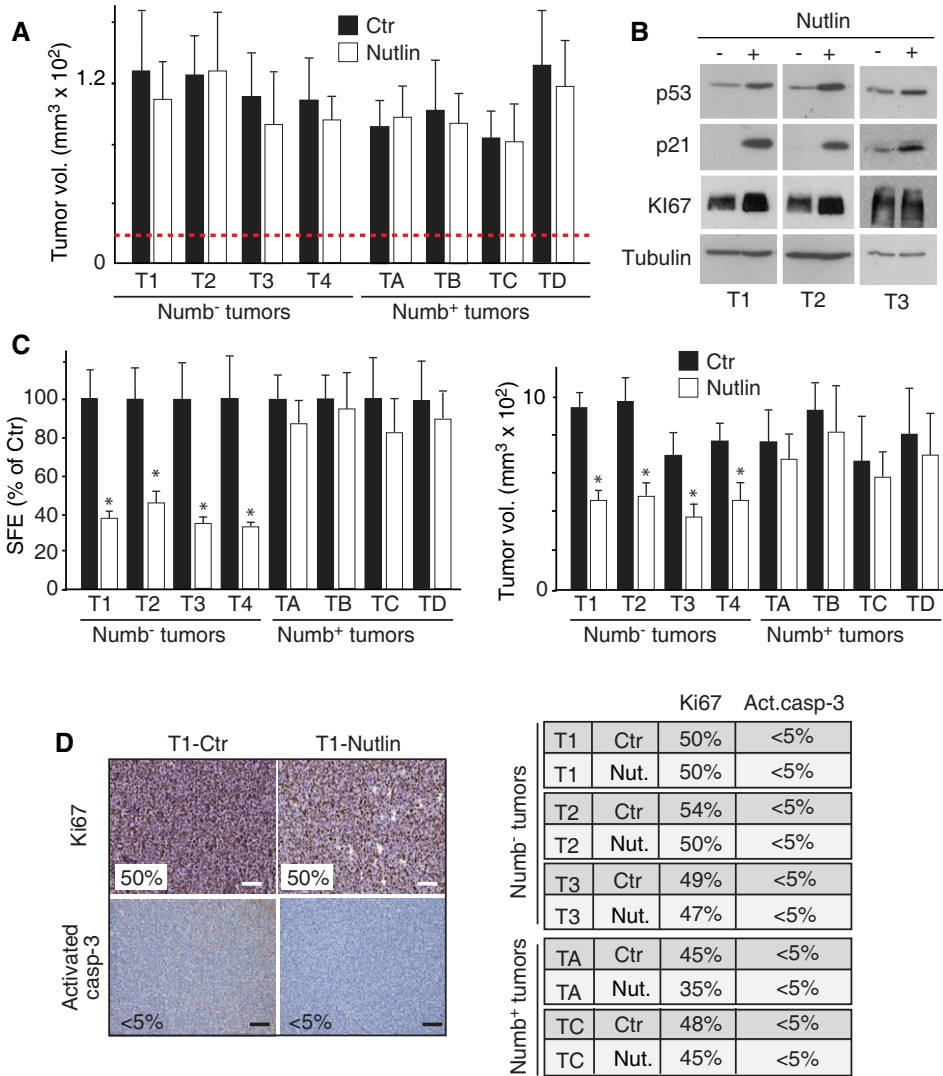

**Figure 6. Effects of *in vivo* Nutlin-3 treatment on CSC content of Numb⁻ and Numb⁺ tumors.**

A    MSs derived from Numb⁻ and Numb⁺ tumors were orthotopically transplanted into NGS mice and allowed to form tumors of ~20 mm³ (dashed red line). Nutlin-3 (Nutlin) or vehicle DMSO (Ctr) was then administered to the mice every 3 days at 20 mg/kg for a period of 12 days, at the end of which, mice were sacrificed and tumor volume was measured. Bars represent the tumor volume, expressed as the mean value of three independent experiments ($\pm$ SD of a minimum of 11 to a maximum of 16 tumors). Unpaired two-sided Student's *t*-test. *$P < 0.05$ vs. Ctr in the matching condition was considered as significant. No significant differences were observed either in Numb⁻ or Numb⁺ tumors.

B, C    Tumors, obtained in (A) were dissociated and cells were analyzed by IB to verify the efficacy of the *in vivo* Nutlin-3 treatment (B), and then either tested for MS-forming efficiency (SFE), (C, left), or re-transplanted orthotopically in mice for 3 weeks (C, right). In (B), panels shown in the IB were assembled either using lanes from the same blot (splicing out lanes loaded with additional controls or non-relevant samples), or from different blots run and stained simultaneously (see also Materials and Methods). Tubulin, loading control. In (C), SFE data are mean values of three independent experiments ($\pm$ SD of 12 measurements) and are expressed relative to the SFE or MS size in Ctr cells (= 100%). Unpaired two-sided Student's *t*-test. *$P < 0.05$ vs. Ctr in the matching condition was considered as significant ($P$-value SFE: T1, 1.01E-10; T2, 8.54E-09; T3, 7.89E-10; T4, 1.16E-09). Right, bars represent the tumor volume, expressed as the mean value of three independent experiments ($\pm$ SD of a minimum of 11 to a maximum of 16 tumors). Unpaired two-sided Student's *t*-test. *$P < 0.05$ vs. Ctr in the matching condition was considered as significant (T1, $P = 4.70E-18$; T2, $P = 9.27E-15$; T3, $P = 5.57E-09$; T4, $P = 6.94E-09$).

D    IHC analysis of Ki67 and activated caspase-3 in tumors generated by the transplantation of tumor cells from animals treated as described in (A). Top, representative IHC images; bottom, quantification of IHC results (> 30,000 cells counted/sample). Scale bars = 100 µm.

# Discussion

Despite considerable efforts placed in the development of novel cancer drugs, most treatments fail during clinical trials. This failure is in part attributable to: (i) the design of pre-clinical validation studies, mostly based on models (established cell lines and their xenografts) insufficient to predict accurately patients' response to treatment, and (ii) current measures of therapy efficacy based on short-term tumor debulking, thereby ignoring the contribution of persistent CSCs to disease progression (Therasse *et al*, 2000; Liu & Wicha, 2010). Herein, we provide a pre-clinical validation of a strategy that addresses these two issues by using orthotopic PDXs as a

**Table 2.** *In vivo* administration of Nutlin-3 selectively affects the tumorigenicity and CSC content of Numb⁻ tumors, but not of Numb⁺ tumors.

| | Tumor | Treatment | Nr. of cells injected | | | | | | Freq. SCs/CICs | 95% CI | *P*-value |
|---|---|---|---|---|---|---|---|---|---|---|---|
| | | | $10^5$ | $10^4$ | $5 \times 10^3$ | $10^3$ | $10^2$ | $10^1$ | | | |
| | | | Outgrowths/injections | | | | | | | | |
| Numb⁻ tumors | T1 | Ctr | 2/2 | 2/2 | 4/6 | 1/8 | 0/2 | 0/2 | 1:4,660 | 1:2,147–1:10,115 | 0.005 |
| | T1 | Nut. | 2/2 | 3/8 | 1/8 | 0/8 | 0/2 | 0/2 | 1:26,284 | 1:10,794–1:64,008 | |
| | T2 | Ctr | 2/2 | 2/2 | 3/6 | 1/6 | 2/8 | 0/2 | 1:4,202 | 1:1,928–1:9,157 | 0.002 |
| | T2 | Nut. | 2/2 | 4/10 | 1/8 | 0/8 | 0/2 | 0/2 | 1:24,200 | 1:10,660–1:54,939 | |
| | T3 | Ctr | 2/2 | 4/4 | 4/6 | 0/6 | 0/2 | 0/2 | 1:4,743 | 1:2,286–1:9,842 | 0.002 |
| | T3 | Nut. | 2/2 | 2/8 | 1/7 | 0/2 | 0/2 | 0/2 | 1:31,731 | 1:12,082–1:83,335 | |
| | T4 | Ctr | 2/2 | 4/4 | 2/4 | 1/6 | 0/2 | 0/2 | 1:4,758 | 1:2,140–1:10,580 | 0.003 |
| | T4 | Nut. | 2/2 | 3/8 | 0/8 | 0/2 | 0/2 | 0/2 | 1:32,319 | 1:12,467–1:83,786 | |
| Numb⁺ tumors | TC | Ctr | 2/2 | 6/6 | 3/8 | 0/8 | 0/2 | 0/2 | 1:6,991 | 1:3,621–1:13,498 | 0.688 |
| | TC | Nut. | 2/2 | 6/6 | 3/6 | 0/8 | 0/2 | 0/2 | 1:5,698 | 1:2,907–1:11,172 | |
| | TD | Ctr | 2/2 | 4/4 | 3/6 | 1/8 | 0/2 | 0/2 | 1:5,403 | 1:2,618–1:11,151 | 0.770 |
| | TD | Nut. | 2/2 | 4/4 | 3/6 | 2/8 | 0/2 | 0/2 | 1:4,630 | 1:2,277–1:9,414 | |

Limiting dilution orthotopic re-transplantation of explanted cells from the indicated tumors treated *in vivo* with Nutlin-3 (Nut.) or vehicle DMSO (Ctr) as in Fig 6A. The complete statistical analysis of the experiment, with the frequency of CICs and with 95% confidence intervals (CI), is shown. Analyses were performed as described in the legend to Table 1.

model system, and by evaluating treatment efficacy based on CSC-specific parameters.

We demonstrated that re-expression of Numb in Numb⁻ BC cells: (i) decreases their MS-forming ability; (ii) reverses their unlimited self-renewal potential and restores self-extinguishing kinetics in the serial MS propagation assay; (iii) decreases their *in vivo* tumorigenic potential; (iv) decreases the size of the CSC pool. In addition, the phenotypes associated with loss of Numb expression were all reverted by restoration of physiological levels of p53 by Nutlin-3. This latter finding is particularly pertinent to BC, where p53 is relatively less frequently mutated compared with other types of tumors, and loss of WT p53 activity may depend on alterations of several upstream regulatory pathways (Gasco *et al*, 2002; Dumay *et al*, 2013). Thus, Numb⁻ tumors behave functionally as p53-null tumors, offering the possibility of combating Numb dysfunction, and its associated cancer phenotypes, downstream at the level of p53.

The deregulation of the Numb/p53 axis is not the sole mechanism through which expansion of the SC compartment and emergence of CSCs can be achieved in BC. Indeed, an amplified SC pool was also evident in Numb⁺ tumors. However, these tumors were insensitive to Numb expression or Nutlin-3 treatment. These data indicate that loss of Numb is a causal event in the subgroup of Numb⁻ BCs, and establish Numb as an authentic tumor suppressor gene in naturally occurring cancers.

The impact of loss of Numb expression on mammary carcinogenesis could be even more far reaching, since this event also causes unchecked Notch activity, which is relevant to the transformed phenotype of Numb⁻ human tumors (Pece *et al*, 2004; Colaluca *et al*, 2008; Westhoff *et al*, 2009). Thus, deregulation of Notch activity might contribute to transformation in Numb⁻ tumors, a notion supported by evidence implicating Notch in the regulation of mammary SC self-renewal and in the proliferative/differentiative balance in progenitors (Weijzen *et al*, 2002; Raouf *et al*, 2008).

Our most significant finding was that in a pre-clinical PDX model of Numb⁻ human BCs, Nutlin-3 treatment behaved as a CSC-specific therapy that efficiently curbs the tumorigenic and metastatic potential of CSCs. We tailored the administration regimen (20 mg/kg injected intraperitoneally every 3 days for a total of 12 days) to minimize the apoptotic and necrotic effects associated with higher concentrations and/or prolonged exposure to the drug (Kunkele *et al*, 2012). In this regard, we note that our Nutlin-3 administration protocol did not cause any significant effect on the growth rate of the primary xenograft (Figs 6A and 7, left), a finding in keeping with reports demonstrating minimal anti-tumoral effects of short-term (1–2 weeks) Nutlin-3 treatments (Tovar *et al*, 2006; Vaseva *et al*, 2011; Kunkele *et al*, 2012). Despite the apparent lack of effect on the primary PDX growth, the use of assays to measure CSC activity (Gupta *et al*, 2009), such as the ability of primary PDX-derived cells to form tumorspheres *in vitro* or to sustain secondary PDX growth following re-transplantation, was able to unmask a specific anti-CSC effect of the drug (Figs 6C and 7, middle/right).

These results are important in light of increasing concerns that the primary endpoint to predict the clinical benefit of cancer therapeutics in clinical trials, that is, the objective tumor reduction according to RECIST (Response Evaluation Criteria in Solid Tumors) criteria (Therasse *et al*, 2000), might underestimate the effects of compounds with anti-CSC action (Liu & Wicha, 2010). Indeed, while the degree of tumor shrinkage measures effects on the bulk tumor cell population, selective targeting of the CSC pool might not result in whole tumor size reduction assessed by the conventional RECIST system. Our results provide evidence of such a scenario and corroborate the idea that, as increasing efforts will be directed to the specific targeting of the CSC compartment in tumors, the design of Phase II trials should be overhauled and integrated with additional biological endpoints based on the CSC paradigm.

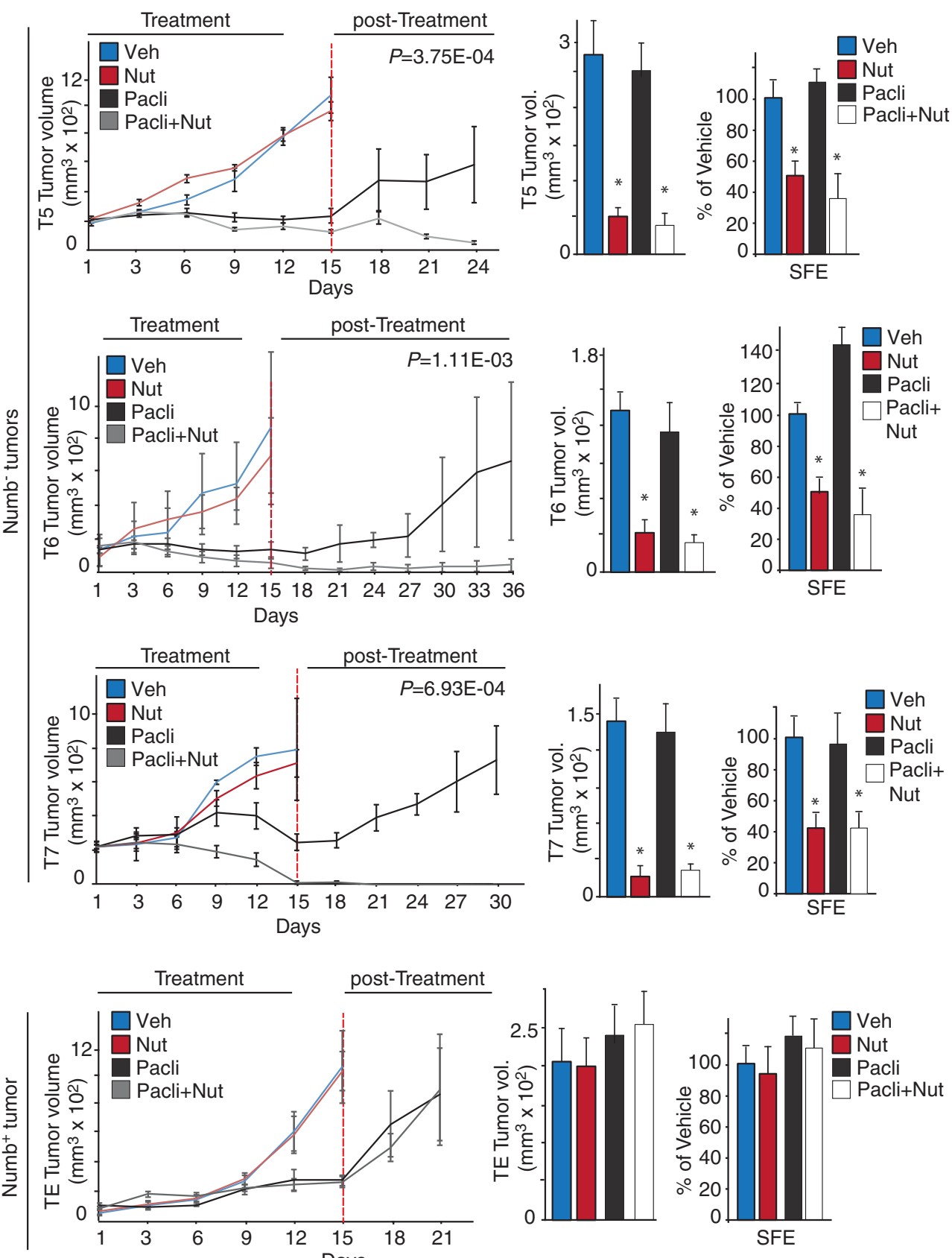

**Figure 7.**

**Figure 7.** *In vivo* **efficacy of Nutlin-3 and paclitaxel used as a monotherapies or in a combinatorial regimen.**
Left panels: MECs derived from three Numb⁻ (T5, T6, T7) and one Numb⁺ (TE) tumors were orthotopically transplanted into NGS mice and allowed to form tumors of ~100–200 mm³. Mice were then randomly assigned to four groups of treatments (3–16 tumors per group): vehicle (Veh), Nutlin-3 (Nut), paclitaxel (Pacli), paclitaxel + Nutlin-3 (Pacli + Nut). Nutlin-3 (20 mg/kg dosed every 3 days for 4 doses) and paclitaxel (20 mg/kg dosed every 6 days for 3 doses) were administered for a period of 12 days (Treatment). At day 15 (dashed red line), mice were sacrificed and explanted tumors (n ≥ 3 tumors/group) were pooled, digested, and used for *in vivo* tumorigenicity (middle panels) or SFE (right panels) assays. Tumor regrowth was monitored following therapy discontinuation in mice (n ≥ 3 mice/group) treated with paclitaxel alone or in combination with Nutlin-3 (post-treatment). Points of the growth curves represent the tumor volume expressed as the mean value of a single experiment (± SE of a minimum of three to a maximum of 16 tumors). An unpaired two-sided Student's *t*-test was used to analyze the degree of tumor shrinkage (expressed as %) in Pacli + Nut- vs. Pacli-treated tumors at day 15 (T5, 33%, *P* = 0.048; T6, 60%, *P* = 0.228; T7, 94%, *P* = 0.022). A paired two-sided Student's *t*-test was used to evaluate the overall tumor reduction (measured at the end of post-treatment) in Pacli + Nut-treated tumors vs. baseline (T5, 72%, *P* = 3.75E-04; T6, 71%, *P* = 1.11E-03; T7, 100%, *P* = 6.93E-04). Middle panels: Cells from pooled tumors (minimum three tumors per condition) explanted at day 15 were assessed for tumorigenicity (see Materials and Methods for experimental details). Bars represent the tumor volume expressed as the mean value of a single experiment (± SE of a minimum of three to a maximum of eight tumors). Unpaired two-sided Student's *t*-test. *\*P* < 0.05 vs. matching condition was considered as significant (*P*, Nut vs. Veh: T5 = 2.0E-04; T6 = 3.20E-04; T7 = 0.017; *P*, Pacli + Nut vs. Pacli: T5 = 6.37E-06; T6 = 6.50E-03; T7 = 5.40E-03). Right panels: SFE data are mean values of a single experiment (± SD of 12 measurements) and are expressed relative to the SFE or MS size in control (Veh) cells (= 100%). Unpaired two-sided Student's *t*-test. *\*P* < 0.05 vs. matching condition was considered as significant (*P*, Nut vs. Veh: T5 = 3.0E-07; T6 = 1.10E-07; T7 = 2.60E-06; *P*, Pacli + Nut vs. Pacli: T5 = 1.80E-12; T6 = 9.70E-12; T7 = 2.10E-05).

Relevant to this point, we also provide preclinical evidence that combining the anti-CSC-specific activity of Nutlin-3 with a standard-of-care chemotherapeutic agent, paclitaxel, enhances the therapeutic response of Numb⁻ PDXs and overcomes drug resistance to chemotherapy treatment. Extrapolated to the real clinical setting of Numb⁻ human BCs, our results argue that the combination of the anti-CSC effects of Nutlin-3 with standard tumor-debulking treatments might represent a promising therapeutic strategy.

Major efforts are being directed toward the restoration of p53 function in human cancers through the inhibition of Mdm2 (Burgess *et al*, 2016). Indeed, several anti-Mdm2 compounds, related to Nutlin-3, are currently in clinical development. However, clinical trials with these compounds face major challenges, including the lack of clear biomarkers for patient selection and the frequent occurrence of adverse effects (Ray-Coquard *et al*, 2012; Andreeff *et al*, 2016). Our results might help circumventing these problems. First, at least in the models herein described based on the use of Numb⁻ and Numb⁺ BCs with a comparable WT-p53 status, the downregulation of WT p53, which occurs in Numb⁻ but not in Numb⁺ BCs, was necessary for the anti-CSC effect of Nutlin-3. Thus, we submit that the presence of WT p53 might not be a sufficient criterion for eligibility to anti-Mdm2 therapies. Rather, patients should be stratified for low levels of WT p53, either identified directly (something that might prove technically challenging) or through surrogate markers such as Numb status.

In addition, we note that the tailoring of the best regimen of administration of anti-Mdm2 compounds might greatly benefit from close-to-reality preclinical models that allow for accurate evaluation of on-target and off-target effects of these drugs. The Numb⁻ BC PDXs might represent one such model. These tumors were responsive to Nutlin-3 treatment under conditions that would have escaped assessment of efficacy in mono-therapy, by RECIST criteria, since the drug affected CSCs in the absence of necrotic/apoptotic effects in the bulk tumor mass and in the absence of tumor shrinkage. We do not know whether the absence of "toxic" effects on the tumor would translate also in absent/reduced systemic toxicity for the patient, at the doses of Nutlin-3 herein employed. If so, however, Numb⁻ BC PDXs might help in identifying the minimal efficacious dose of Nutlin-3, or of other anti-Mdm2 compounds, to obtain efficient anti-CSC targeting, while reducing general toxicity: a possibility that warrants further investigations.

# Materials and Methods

### Clinical samples

Fresh, frozen, or archival formalin-fixed paraffin-embedded (FFPE) mammary tissue specimens were collected at the European Institute of Oncology (IEO, Milan, Italy). All tissues were collected via standard operating procedures approved by the Institutional Ethical Board, and informed consent was obtained for all tissue specimens linked with clinical data.

All BCs employed in this study possessed WT p53, assessed by sequencing of the p53 coding sequence. Briefly, the human *P53* gene (NM_000546.5, 10 coding exons) was amplified by PCR starting from genomic DNA extracted from FFPE samples. We designed short amplicons (maximum size ~200 bp) to avoid possible problems due to degradation/fragmentation of DNA extracted from FFPE samples. Since large exons were divided into more overlapping amplicons, the entire p53 coding sequence was covered by 13 amplicons (primers available upon request). Each forward and reverse primer was designed with a 5′ universal tail (PE21 forward; M13 reverse), which was subsequently used to directly sequence the PCR fragments using BigDye v3.1 chemistry from Applied Biosystems. The sequencing reactions were analyzed with the Mutation Surveyor software (SoftGenetics).

The Numb status was attributed to the tumors by measuring the levels of Numb expression by IHC. Normal mammary tissues displayed homogenous and intense Numb staining in the luminal layers (IHC score 3) (see also Pece *et al*, 2004; Colaluca *et al*, 2008). Tumors were classified on an IHC scale from 0 to 3 (0 = undetectable expression, 3 = expression comparable to that of normal luminal mammary cells). Tumors were classified as Numb⁻ if > 70% of the cells displayed a score ≤ 1, or as Numb⁺ if > 60% of the cells displayed a score of 3.

### *In vivo* studies using PDXs of Numb⁻ and Numb⁺ human breast cancers

To generate PDX models, fresh specimens from Numb⁻ and Numb⁺ human BCs were cut into 4 × 2 mm pieces using a razor blade, removing necrotic tissue, if present. The resulting fragments were embedded in ice-cold Matrigel (BD Matrigel™, BD Biosciences) and immediately injected into the 4th inguinal mammary glands of 8-week-old female

NOD/SCID/IL2Rγ$^{-/-}$ (NSG) mice (Jackson Laboratories stock #5557) anesthetized by i.p. injection of 150 mg/kg tribromoethanol (Avertin). Animals were euthanized when the tumor exceeded 10% of the body mass in compliance with regulations for use of vertebrate animals in research. The tumor volume was determined by measuring the tumor in two dimensions and calculated using the mathematical formula for a prolate ellipsoid: tumor volume = $(L \times W^2)/2$, where L = length of the longest diameter; W = length of the shortest diameter. Female age-matched animals were assigned randomly to the treatment (Numb overexpression or Nutlin-3) or control (control vector or vehicle) groups. Mice were monitored twice weekly by an investigator blinded to treatment assignment. No animals were excluded from the analysis. The size of the animal cohort was established with the aim to minimize the number of animals used considering that inter-animal variations in the mammary fat pad engraftment is generally low starting from fixed amounts of primary cells under standardized transplantation conditions.

Note that since tumor biopsy specimens are a limited source of material and primary MECs can be propagated in culture only for a limited number of generations, not all experiments could be performed on the entire set of BCs. In all critical experiments, we used a minimum of two Numb$^-$ and one Numb$^+$ BCs in each case, and frequently more.

For *in vivo* limiting dilution transplantation studies, decreasing concentrations of cells dissociated from Numb$^-$ and Numb$^+$ PDXs were resuspended in a 1:1 volume of PBS/Matrigel and immediately transplanted into the abdominal mammary gland of 8-week-old female NSG mice. The transplantation frequency was calculated by Poisson statistics, using the "StatMod" software package for the R computing environment (http://www.R-project.org), as previously described (Shackleton *et al*, 2006). A complementary log-log generalized linear model was used: two-sided 95% Wald confidence intervals or, in case of zero outgrowths, one-sided 95% Clopper–Pearson intervals. The single-hit assumption was tested as recommended and was not rejected for any dilution series ($P > 0.05$). All *in vivo* experiments were approved by the Italian Ministry of Health and performed in accordance with the Italian laws (D.L.vo 116/92 and following additions), which enforce the EU 86/609 directive, and under the control of the institutional organism for animal welfare and ethical approach to animals in experimental procedures (Cogentech OPBA).

To test the *in vivo* efficacy of Nutlin and paclitaxel alone or in combination regimens, cells dissociated from Numb$^-$ and Numb$^+$ PDXs were resuspended in a 1:1 volume of PBS/Matrigel and injected into the fat pad of 8-week-old NSG mice. Mice were randomly assigned into the different treatment groups (from a minimum of 3 to 8 mice per group) when tumors reached approximately 100–200 mm$^3$. Animals received intraperitoneal injection of either vehicle drug (10% DMSO, 10% polysorbate 80 in PBS) or Nutlin-3, paclitaxel, or a combination of Nutlin-3 plus paclitaxel. Nutlin-3 was administered at 20 mg/kg dosed every 3 days (for a total of 4 doses) and paclitaxel at 20 mg/kg dosed every 6 days (for a total of 3 doses) over a period of 12 days. Changes in tumor burden were assessed every 3 days with a caliper. All the tumor size measurements were done by two investigators, with one investigator blinded to the treatment group. For the studies performed using the tail vein model of breast cancer lung metastasis, MECs dissociated from MSs derived from two Numb$^-$ and two Numb$^+$ primary tumors were resuspended in 100 μl saline and injected via lateral tail vein in 7- to 8-week-old NSG mice. Two hundred and fifty thousand cells, treated either with Nutlin-3 or vehicle as a control, were injected for each Numb$^-$ or Numb$^+$ tumor, using five mice for each treatment group. Quantitative analysis of the lung metastatic tumor burden was performed after 3 weeks post-injection.

## Cells, reagents, and immunoblotting

Bulk mammary epithelial cells (MECs) were isolated from human mammary tumor samples or from PDXs, and cultivated in suspension culture to yield first (F1) or second (F2) generation mammospheres (MSs), as previously described (Pece *et al*, 2010; Tosoni *et al*, 2012, 2015). Briefly, tumor biopsies or PDXs were mechanically dissociated and enzymatically digested in DMEM/F12 medium supplemented with 1 mM glutamine, 200 U/ml collagenase (Sigma), and 100 U/ml hyaluronidase (Sigma) at 37°C for 4–5 h under rotating conditions. Single cell suspensions were obtained through an additional incubation of digested tissues in 0.05% trypsin–EDTA for 5 min at 37°C.

Nutlin-3 was either purchased from Cayman Chemical or generously supplied by S. Minucci and M. Varasi (Drug Discovery Unit, European Institute of Oncology, Milan, Italy). The (+)-enantiomer Nutlin-3b was from Cayman Chemical. Cisplatin was from TEVA ITALIA. MG132 was from Enzo Life Science. Paclitaxel (Taxol, 6 mg/ml) was from Bristol-Myers Squibb.

Antibodies for immunoblotting (IB) were directed against: tubulin or vinculin (Sigma); Numb (AB21, a mouse monoclonal Ab against a.a. 537–551 of hNumb; Colaluca *et al*, 2008); p53 (12C2) and activated caspase-3 (Asp175)—Cell Signaling Technologies; p21 (F-5; SantaCruz Biotechnology); GRP94 (9E10; Enzo Life Sciences); β-catenin (BD Biosciences); Ki67 (SP6; Thermo Scientific Lab Vision). Panels shown in the various IBs were assembled either using lanes from the same blot (splicing out lanes loaded with additional controls or non-relevant samples), or from different blots run and stained simultaneously. In some cases, given the nature of the starting materials (primary cultures or MS established immediately after surgery that do not last long in culture), samples were run and immunoblotted at different times. When these samples are displayed comparatively, they are derived from gels in which internal controls (spliced out in the presentations) were also loaded to allow for comparison. In all cases, the splicings are clearly marked with solid bars or separated by white lines. Details of primary antibodies and procedures used for flow cytometry studies can be found in Appendix Supplementary Methods.

## Engineering of vectors, shRNA experiments, and qPCR

The lentiviral construct harboring the Numb-DsRed and the Numb-GFP fusion proteins and the procedures for lentiviral infection have been previously described (Pece *et al*, 2010; Tosoni *et al*, 2015). MSs from Numb$^-$ and Numb$^+$ human BCs were dissociated, lentivirally transduced with Numb-DsRed or with the control pLVX vector, and cultured in non-adherent conditions for 7 days.

qRT–PCR analysis of cells from dissociated MSs was performed using the TaqMan Cells-to-CT kit (Ambion), which allows gene expression analysis from limited numbers of cells. Each sample was tested in triplicate. For quantitation of gene expression changes in each sample, the ΔCt method was used to calculate relative fold

changes normalized against two different housekeeping genes. TaqMan Gene Expression Assay IDs (Applied Biosystems, CA) were as follows: Hs00355782-m1 (*CDKN1A*, NM_078467), Hs00234753-m1 (*MDM2*, NM_002392), Hs02387400-g1 (*NANOG*, NM_024865.2), Hs99999905-m1 (*GAPDH*), Hs99999903-m1 (*ACTB*).

## Imaging studies

Time-lapse video microscopy was performed with a Scan^R screening station (Olympus-SIS) equipped with a microscope incubation chamber (Evotec). Cells from dissociated normal or tumor MSs were re-suspended in methylcellulose in complete medium, plated onto glass bottom dishes, and observed through a 10× 0.4 NA objective. Both DIC and DsRed-epifluorescence images were collected with auto-focusing procedures and compensated for focal shift. Different focal planes were recorded to prevent loss of image contrast due to axial cell movement. Images were captured every hour for 7 days, starting 16–24 h after plating, and were reconstructed using the ImageJ software. Confocal analyses were performed with a Leica TCS SP2 AOBS microscope (see also Appendix Fig S2 for additional experimental descriptions).

For the experiments described in Appendix Fig S2C, MECs from the N1 specimen (normal counterpart of tumor T1) were lentivirally transduced with Numb-GFP in adherent conditions and stained with PKH26 (Sigma) before seeding them in suspension to allow MS formation. Imaging was as above.

For IHC analysis, 3-μm-thick sections of FFPE primary tumors or PDXs were assayed with the appropriate antibodies (Numb, clone C29G11 and anti-activated caspase-3, clone Asp175 from Cell Signaling Technologies; Ki67, clone SP6 from Thermo Scientific Lab Vision). Slides were digitally scanned with the Aperio ScanScope XT. For the quantitation of Ki67 and activated caspase-3 stainings, the slides were automatically analyzed with the Aperio ImageScope IHC algorithm (Aperio Technologies, Inc, Vista, CA, USA). Areas containing 30–60,000 cells were analyzed, and the percentage of positive cells was calculated. Digital images were processed with Adobe Photoshop CS3.

**Expanded View** for this article is available online.

## Acknowledgements
We thank the anonymous patients who donated their samples for research. We also thank S. Minucci and M. Varasi for Nutlin-3; the IEO Pharmacy for pacli-taxel; J. Quarna, M. Coazzoli and C. E. Villa for technical assistance; the Veterinary Facility, the DNA Sequencing Unit, the IEO Biobank, the IEO Imaging Service and the Molecular Pathology of the IEO Molecular Medicine Program; R. Gunby for critically editing the manuscript. This work was supported by grants from the Associazione Italiana per la Ricerca sul Cancro (AIRC—IG 11904 to SP; IG 18988 and 14404 to PPDF and MCO 10.000), MIUR (the Italian Ministry of University and Scientific Research), the Italian Ministry of Health to SP, PPDF, and DT; the European Research Council (Mammastem Project), the Monzino Foundation to PPDF.

## Author contributions
DT, SP, BES, and SZ performed experimental work and analyzed data. GB, GP, and GV provided samples and supervised the histopathological analysis. PPDF and SP planned and supervised the project, performed data analysis, and wrote the manuscript.

**The paper explained**

**Problem**
In many types of cancers, including breast cancer (BC), the intrinsic refractoriness of cancer stem cells (CSC) to conventional anti-cancer therapies is though to underlay therapy failure and disease progression. Thus, the elucidation of the molecular mechanisms responsible for the emergence and maintenance of CSCs holds great promise toward the development of targeted anti-CSC therapies.

**Results**
Using patient-derived xenografts (PDX), as a clinically relevant model of Numb-deficient breast cancers (BC), we show that the unlimited self-renewal and high tumorigenic potential of CSCs in Numb-deficient BCs can be selectively reverted by re-expression of the tumor suppressor Numb, or pharmacological restoration of p53 function with the Mdm2 inhibitor Nutlin-3. We further show that targeting the Numb/p53 dysfunction selectively interferes with the CSC compartment of Numb-deficient BCs, with only modest, if any, effects at the level of the bulk tumor population. The combined use of Nutlin-3 with standard chemotherapy increases the response to therapy of Numb-deficient BCs.

**Impact**
Our work provides evidence of a molecular mechanism responsible for the emergence and maintenance of CSCs in Numb-deficient BCs, which represent a sizable fraction of naturally occurring and intrinsically aggressive human BCs. Through the use of orthotopic PDX and specific assays to study CSC biology, we provide proof-of-evidence that targeting the Numb/p53 dysfunction constitutes an effective anti-CSC therapy. We also provide preclinical evidence that a selective anti-CSC therapy is feasible, based on the elucidation of the mechanisms underlying the emergence and the expansion of CSCs. Extrapolated to the real clinical setting of human BCs, our results suggest that Numb represents a suitable predictive biomarker for eligibility to therapies able to restore p53 function, such as the Nutlin-related class of anti-Mdm2 inhibitors. Furthermore, based on evidence that targeting p53 dysfunction potentiates the response of Numb⁻ BCs to chemotherapy, our results also pave the way for a combinatorial treatment of these tumors.

## Conflict of interest
The authors declare that they have no conflict of interest.

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
