## [Review Process File · EMBO Molecular Medicine]

Pre-clinical validation of a selective anti-cancer stem cell therapy for Numb-deficient human breast cancers

Daniela Tosoni, Sarah Pambianco, Blanche Ekalle Soppo, Silvia Zecchini, Giovanni Bertalot1, Giancarlo Pruneri, Giuseppe Viale, Pier Paolo Di Fiore and Salvatore Pece

Corresponding author: Pier Paolo Di Fiore and Salvatore Pece, European Institute of Oncology

Review timeline:

Submission date:	12 August 2016
Editorial Decision:	14 September 2016
Revision received:	24 January 2017
Editorial Decision:	06 February 2017
Revision received:	12 February 2017

Transaction Report:

Editor: Roberto Buccione

1st Editorial Decision

14 September 2016

Thank you for the submission of your manuscript to EMBO Molecular Medicine. We are very sorry that it has taken longer than usual to get back to you on your manuscript.

As I had anticipated, due to the holiday season we experienced unusual difficulties in securing three willing and appropriate reviewers. As a further delay cannot be justified I have decided to proceed based on the two available and consistent evaluations.

As you will see, the reviewers clearly find your work both interesting and important. However they also clearly agree that a fundamental issue must be resolved to upgrade the pre-clinical relevance and impact of the work presented in this manuscript. Namely, pre-clinical evidence must be provided that a Nutlin/chemotherapy combination can indeed target both the bulk and tumour initiating cell components (if such is the case), thus effectively potentiating chemotherapy and circumventing TIC-related drug resistance. I should add that I had similar concerns on the study when deciding whether to send the manuscript out for peer-review. Reviewer 2 also mentions a few other issues that require your action.

In conclusion, while publication of the paper cannot be considered at this stage, given the potential interest of your findings and after internal discussion, we have decided to give you the opportunity to address the criticisms. Please consider that the concerns raised are of great importance for us as they impinge on the overall quality and robustness of experimental support for the main conclusions.

We are thus prepared to consider a substantially revised submission, with the understanding that the reviewers' concerns must be addressed with additional experimental data where appropriate and that

acceptance of the manuscript will entail a second round of review.

I look forward to seeing a revised form of your manuscript as soon as possible.

***** Reviewer's comments *****

Referee #1 (Comments on Novelty/Model System):

In this nice paper from Pece and coworkers it is shown that loss of Numb results in enhanced BCa stem cell (Csc) activity due to down regulation of p53. Re expression on Numb reduces Csc activity through up regulation of p53. The same can be achieved via Nutlin3 treatment. Since the latter only affects Csc it does not lead to tumor shrinkage. While the work is solid and convincing, I suggest that the authors demonstrate the value of Nutlin therapy by showing its ability to potentiate cytotoxic chemotherapy and prevent acquired drug resistance which presumably depends on Csc.

Referee #2 (Remarks):

In the manuscript by Daniela Tosoni et al., the authors investigate the potential of Nutlin-3 treatment as a selective anti-cancer stem cell therapy for Numb- breast tumors. In previous reports the authors already demonstrated that Numb play a central role in the maintenance of the cancer stem cell (CSC) pool. They also reported that numb control CSCs-fate through the regulation of a Numb/P53/MDM2 axis. Thus, this work is the follow-up of all these previous studies. The authors now propose to transfer these fundamental data into clinics with this preclinical study. In general, this is an interesting manuscript with several interesting observations. However, several additional experiments/controls would be needed to support the conclusions that the authors try to draw in this manuscript.

Major comments:

- 1: One of the biggest concern is, that in all pre-clinical validation the experimental compound (here nutlin-3a) have to be compared to the reference treatment. It is well known that CSCs resist to chemotherapies but as mentioned by the authors CSC-specific therapy have limited effect on tumor growth and will need to be combine with chemotherapy in clinics. Thus it is crucial to know if a combination of nutlin-3a with chemotherapy is able to efficiently target the bulk and the CSC population.
- 2: CSCs are known to generate metastasis. One additional proof of the anti-CSC potential of Nutlin3a treatment will be to detect a diminution of the metastatic burden. Do these PDX models generate metastasis?
- 3: It will very informative to evaluate the evolution of the classical CSC markers (CD44/CD24, ALDH) before and after treatment. In a clinical trial it may serve as surrogate markers of CSC evolution.
4. To compare the effect on the tumor growth, the authors should provide growth curves.

Minor points:

1. Exact p-value should be added to Table 1 and 2
2. Last sentence of the discussion is clearly overstated. Several anti-CSC targeted therapies in human solid tumors have been reported. Some clinical trials have already started (Kaiser J, Science 2015)

1st Revision - authors' response

24 January 2017

Referee #1

General comment:

In this nice paper from Pece and coworkers it is shown that loss of Numb results in enhanced BCa stem cell (Csc) activity due to down regulation of p53. Re expression on Numb reduces Csc activity through up regulation of p53. The same can be achieved via Nutlin3 treatment. Since the latter only affects Csc it does not lead to tumor shrinkage. While the work is solid and convincing, I suggest

that the authors demonstrate the value of Nutlin therapy by showing its ability to potentiate cytotoxic chemotherapy and prevent acquired drug resistance which presumably depends on Csc.

Agree. We thank the reviewer for the appreciative words on our study. We have performed the requested experiment, which is similar to a request from Reviewer #2. Thus, we kindly refer this Reviewer to the reply to point 1 of Reviewer #2 below.

Referee #2

Major comments:

1: One of the biggest concern is, that in all pre-clinical validation the experimental compound (here nutlin-3a) have to be compared to the reference treatment. It is well known that CSCs resist to chemotherapies but as mentioned by the authors CSCspecific therapy have limited effect on tumor growth and will need to be combine with chemotherapy in clinics. Thus it is crucial to know if a combination of nutlin-3a with chemotherapy is able to efficiently target the bulk and the CSC population.

Agree. This is an important point that was also raised by Reviewer #1. To address this point, we have performed *in vivo* efficacy studies by testing Nutlin-3 and a standard-of-care chemotherapy drug, Paclitaxel, alone or in combination. The results are shown in the new Fig. 7, and they demonstrate that:

- i) Nutlin-3 potentiates the anti-tumoral effects of Paclitaxel, inducing a persistent tumor response associated with varying degrees of overall tumor shrinkage, up to complete tumor regression.
- ii) Nutlin-3 overcomes drug resistance by preventing re-initiation of tumor growth after chemotherapy discontinuation.

2: CSCs are known to generate metastasis. One additional proof of the anti-CSC potential of Nutlin3a treatment will be to detect a diminution of the metastatic burden. Do these PDX models generate metastasis?

Agree. Our preclinical breast cancer PDX model does not allow us to follow spontaneous metastasis, due to the excessive size reached by tumors in a timeframe that is comparatively shorter with that required for spontaneous metastasis. We have therefore addressed this important question using the tail vein injection model of breast cancer lung metastasis. Results are depicted in the new Fig. 5 and show a significant impairment of the metastatic potential in Numbdefective, but not in Numb-proficient, cells pre-treated *in vitro* with Nutlin-3.

3: It will very informative to evaluate the evolution of the classical CSC markers (CD44/CD24, ALDH) before and after treatment. In a clinical trial it may serve as surrogate markers of CSC evolution.

Agree. This is an interesting point. We have performed cytofluorimetry experiments, for three PDXs for which enough cells were available, in the various conditions of treatment reported in the new Fig. 7. Overall, the results support the conclusion that Nutlin-3 selectively affects the CSC compartment. The data are now appended to this letter for the reviewer's perusal.

Our feeling is that it would be premature to add these data to the paper, as they would require much more work to reach robust conclusions. Among our reasons of concern:

1. The fact that there is no agreement in the literature as to which marker configuration is optimal for identification of CSCs.
2. The fact that the two configurations that we used (CD44/CD24 and ALDH status) detect different percentages of putative CSCs in the same sample.
3. The fact that the molecular characteristics of the tumor impact on the marker profile: see for instance T7 in the addendum below, in which we could not find detectable levels of CD44+/CD24-cells, a finding that can be explained by the luminal status of this tumor.

For all these reasons, we would be inclined not to include the results in the paper. However, we are ready to do so, if the reviewer deems it necessary. In that case, we will add them as supplemental information, and explicitly mention all the connected caveats.

4. To compare the effect on the tumor growth, the authors should provide growth curves.

Agree. We have followed the approach suggested by this Reviewer in the *in vivo* efficacy experiments described in the new Fig. 7

Minor points:

1. Exact *p*-value should be added to Table 1 and 2.

Agree. The exact *P* values have now been included in Table 1 and 2.

2. Last sentence of the discussion is clearly overstated. Several anti-CSC targeted therapies in human solid tumors have been reported. Some clinical trials have already started (Kaiser J, Science 2015).

Agree. We have revised the text in accordance with the Reviewer's suggestion.

Material and Methods for experiments described in the Addendum Figure

At the end of the treatment with Nutlin-3 or Paclitaxel alone, or with a combination of Nutlin-3+Paclitaxel (same setting of the *in vivo* efficacy studies described in the new Fig. 7), randomly selected tumors explanted from Numb- and Numb+ tumor-bearing mice were digested and the resulting cell populations were analyzed for *in vivo* tumorigenicity or sphere-forming efficiency, as functional assays for CSC activity, or subjected to FACS analysis to measure the distribution pattern of CD44+/CD24- and ALDH-positive cells, in the absence of any further treatment. For FACS analysis, cells were subjected to Aldefluor assays and co-stained with APC-CD44 and PE-CD24 antibodies, or stained with Dye eFluor(R)450 (eBioscience) to assess cell viability. Only lived cells were analyzed. The Aldefluor assay was performed using the manufacturer's recommended protocol (ALDEFLUOR kit, StemCell Technologies, Durham, NC, USA). Briefly, one-million single cell suspensions were incubated in Aldefluor buffer the ALDH protein substrate (BAAA, BODIPY-aminoacetaldehyde, 1 mmol/L) for 45 minutes at 37°C. Cells that could catalyze BAAA to its fluorescent product (BAA) were considered ALDH+. Sorting gates for FACS were drawn relative to cell baseline fluorescence, which was determined by the addition of the ALDH-specific inhibitor diethylaminobenzaldehyde (DEAB) during the incubation. For CD44 and CD24 stainings, cells were resuspended in a 100 µL staining volume of FACS buffer (HBSS + 3% bovine serum albumin), incubated on ice with CD44-APC (BD Biosciences) or CD24-PE (BD Biosciences), according to the manufacturer's recommended protocol, and kept on ice for 40 min. The Dye eFluor(R)450 (eBioscience) or DAPI was used to measure viable, apoptotic and dead cells that were excluded from analysis. Duplicates and dead cells were also excluded by gating with FSC and SSC. Cells were analyzed in a FACS Attune (Life Technologies) and the acquisition and analysis software were Attune Nxt 2.5 and Kaluza analysis 1.5A. Sorting gates for FACS were drawn relative to cell baseline fluorescence of isotype controls.

ADDENDUM

Nutlin-3 used alone or in combination with Paclitaxel affects the expression of the breast cancer stem cell markers CD44, CD24 and ALDH.

Thank you for the submission of your revised manuscript to EMBO Molecular Medicine.

We have now received the enclosed report from reviewer 1, who was asked to re-assess it. As you will see the reviewer is now globally supportive but would like you to include the addendum data provided in your rebuttal as an additional supplementary (Appendix) figure. I would suggest you do so, while also explaining the limitations of the data. In such case, please make sure that you introduce the appropriate callout in the manuscript where appropriate. Alternatively, since as you know we publish the complete peer-review correspondence alongside the paper (unless the authors opt-out, which occurs very rarely), you could simply leave the data in the rebuttal. The latter however, would be a less elegant solution.

Since reviewer 1 was not immediately available and to avoid further delay, I asked reviewer 2 to also confirm whether your response to reviewer 1 is satisfactory, given that the concerns were overlapping. Reviewer 2 commented that s/he agrees that you also satisfactorily addressed the concerns raised by reviewer 1.

I am thus pleased to inform you that we will be able to accept your manuscript pending the following final amendments:

- 1) Please change the orientation of Fig. 5 from landscape to portrait
- 2) Please separate the western blot rows in figures 4A, 6B and S1B with clear, same-width white lines and explain that the figures are assembled from different blots (if such is the case) in the figure legends; see also point 4 below.

Please submit your revised manuscript within two weeks. I look forward to seeing a revised form of your manuscript as soon as possible.

***** Reviewer's comments *****

Referee #2 (Remarks):

In the revised version by Tosoni et al, the authors did a very good job at addressing all my previous points/concerns. The conclusions are now much better supported by their experiments. I have the feeling that the additional data on the CSC markers (addendum) are informative and must be included in the article as supplementary figure.

2nd Revision - authors' response

12 February 2017

In detail, we have now introduced in the manuscript the following changes:

- 1) We have changed the orientation of Figure 5 from landscape to portrait.
- 2) We have separated western blot rows with white bars in Figures 4A, 6B and S1B, and explained in the respective figure legends the origin of the splicings.
- 3) We have included flow cytometry data on the CD44/CD24 and ALDH cancer stem cell markers in a new Appendix Figure (Appendix Fig. S5) with the appropriate callout in the manuscript, and discussed in the corresponding figure legend the caveats that may potentially limit the clinical value of the cancer stem cell profiles that we used in our studies. We have also included as *Appendix Supplemental Methods* a detailed description of the reagents and procedures used for flow cytometry studies.

Corresponding Author Name: Salvatore Pece

Manuscript Number: EMM-2016-06940